# Breastfeeding Attitudes and Their Associated Factors Among Chinese Nursing Undergraduates: A Cross-Sectional Study

**DOI:** 10.3390/nu17193169

**Published:** 2025-10-08

**Authors:** Han Liu, Yutong Xia, Yuchen Deng, Zhuosen Shang, Xiyang Li, Yalan Gu, Jing Sun, Ying Chen

**Affiliations:** School of Nursing, Anhui Medical University, Hefei 230032, China; liuhan20010411@163.com (H.L.); 18324903757@163.com (Y.X.); m18226920572@163.com (Y.D.); shangzhuosen2024@126.com (Z.S.); 15705207963@163.com (X.L.); 13023007331@163.com (Y.G.)

**Keywords:** breastfeeding, nursing undergraduates, knowledge, attitude, associated factors

## Abstract

Background: Breastfeeding promotion is a public health priority in China, yet the exclusive breastfeeding rate remains below national targets. Nursing students, as future key promoters, often report insufficient knowledge, but their attitudes are less clear. Objective: This study aimed to assess breastfeeding attitudes and identify their associated factors among Chinese nursing undergraduates, thereby providing an evidence base for the design of effective educational interventions. Design, Setting and Participants: A cross-sectional study was conducted from October 2024 to January 2025 at a medical university in Anhui Province, China, with 753 nursing students participating. Methods: The participants completed the General Information Questionnaire, the Chinese version of the Comprehensive Breastfeeding Knowledge Scale (CBKS), and the Iowa Infant Feeding Attitude Scale (IIFAS). We analyzed the data via Spearman correlation, univariate analysis, and multiple linear regression. Results: The overall IIFAS score for nursing students was 54 (51, 59), with attitude scores showing a significant positive correlation with knowledge (r = 0.462, *p* < 0.001). Multiple linear regression revealed that breastfeeding attitudes were significantly predicted by CBKS score (*β* = 2.975), grade (*β* = 2.887), major (*β* = 3.235), and breastfeeding intention (*β* = 8.089, all *p* < 0.001), as well as by feeding type before six months (*β* = −1.591, *p* = 0.020). The overall model accounted for 32.7% of the variance (*R*^2^ = 0.327, *F* = 51.666, *p* < 0.001). Conclusions: This study demonstrates that Chinese nursing undergraduates hold predominantly neutral attitudes toward breastfeeding. These attitudes show significant associations with their knowledge level and personal feeding intention, which underscores the necessity of integrating attitude-focused education into nursing curricula.

## 1. Introduction

Breast milk is the most natural and optimal nutrition for newborns, benefiting babies, mothers, and families [1,2]. Research indicates that the advantages of breastfeeding during infancy can extend into childhood and adulthood [3]. The World Health Organization and the United Nations Children’s Fund recommend that infants should start breastfeeding within one hour after delivery; infants under six months should be exclusively breastfed, then safe and reasonable complementary food should be added; and breastfeeding should continue until the age of two or more [4]. The Global Nutrition Monitoring Framework aims to achieve a global exclusive breastfeeding rate of 50% for infants under 6 months by 2025 [5]. However, China’s current exclusive breastfeeding rate for infants under six months of age is only 29%, which is significantly lower than the global average of 43% [6]. Bridging this gap between China’s breastfeeding rates and the national target necessitates the development of effective strategies to enhance breastfeeding practices.

Recently, a systematic review emphasized the critical role of healthcare professionals in initiating and sustaining exclusive breastfeeding [7]. Research indicates a positive association between healthcare professionals’ support and higher rates of exclusive breastfeeding [8]. Insufficient breastfeeding education for prenatal mothers poses a barrier to postpartum breastfeeding and contributes to early cessation [9,10,11]. Discrepancies in breastfeeding knowledge, attitudes, and skills among healthcare professionals [12,13,14] can result in inadequate dissemination of information, potentially leading to premature discontinuation of breastfeeding [15,16]. Research indicates that one of the important reasons for Chinese health medical professionals’ limited effectiveness in promoting breastfeeding lies in healthcare providers’ lack of essential knowledge and practical skills [12]. Nursing undergraduates, as future healthcare providers, are essential in driving meaningful changes in breastfeeding practices in China. Understanding their attitudes toward breastfeeding is crucial for effective breastfeeding education and promotion efforts.

Although the recognized importance of breastfeeding has many benefits, breastfeeding education can promote breastfeeding practices. However, in China and even worldwide, breastfeeding education is not satisfactory. One study revealed that nursing undergraduates in China have insufficient knowledge of breastfeeding, and there is a decreasing trend in the pertinent knowledge and practice of breastfeeding in China [17]. Moreover, a systematic review of studies in seven countries revealed that nursing students do not always receive adequate breastfeeding training at universities [18]. To promote breastfeeding, strengthening breastfeeding courses in undergraduate nursing education is urgently needed. The Multi-theory Model of Health Behavior Change (MTM) provides a systematic framework for understanding and facilitating behavior modification [19], which is highly relevant to addressing the gaps in healthcare providers’ ability to promote breastfeeding. Specifically, MTM advocates for designing tailored training programs to build healthcare providers’ confidence in addressing common lactation challenges [20]. This recommendation aligns closely with the core objective of our study.

In this study, we investigated the breastfeeding attitudes of nursing undergraduates and the factors that may influence them through a cross-sectional approach. The findings may offer important insights for developing breastfeeding education programs, shaping curricula, and implementing targeted interventions, thereby serving as a reference for enhancing the breastfeeding attitudes of nursing undergraduates. The methodology and identified factors also offer a model that can be replicated and applied to nursing education on a global scale. Concurrently, exploring the factors that facilitate or inhibit breastfeeding attitudes among nursing undergraduates can help identify sociodemographic differences within the student population. This, in turn, enables the design of more precise breastfeeding education programs, ultimately facilitating a tailored approach to instruction.

## 2. Methods

### 2.1. Study Design

This observational, descriptive and cross-sectional study was conducted at a medical university in Anhui Province, China, from October 2024 to January 2025.

### 2.2. Setting and Study Participants

The study population comprised full-time nursing undergraduates across three grades (freshman, sophomore, and junior) and three majors (nursing, midwifery, and elderly service management, three different majors in the faculty of nursing). Sample size determination followed the standard for multiple regression analysis, with a minimum requirement of 20 times the number of independent variables [21]. There are 13 independent variables; considering the nonresponse rate of 20%, the final sample size is at least 325. Ultimately, the study included 753 participants on the basis of actual recruitment outcomes.

### 2.3. Tools of the Study

The data were collected via a self-administered questionnaire comprising three parts.

#### 2.3.1. The General Information Questionnaire

In accordance with the study’s objectives and the literature review, a self-designed questionnaire was utilized to gather general information about the participants (The General Information Questionnaire). This information included socio-demographic characteristics (age, gender, residential location type, only child, grade, major), academic and breastfeeding-related factors (interest in major, completion of mother-infant-related courses, internship experience in obstetrics or neonatology, exposure to breastfeeding advocacy by teachers, intention to breastfeed in the future or support a partner in breastfeeding), and personal experiences (witnessing breastfeeding and own feeding type before six months of age).

#### 2.3.2. Comprehensive Breastfeeding Knowledge Scale (CBKS)

The Chinese version of the CBKS was utilized to assess students’ knowledge of breastfeeding. Authorization for the use of CBKS was obtained from the original developer, Jennifer Abbass-Dick, and the translator of the Chinese version, Qin Zhu. Developed in 2019 by Abbass-Dick [22], CBKS was tailored to meet the educational standards of the Baby-friendly Hospital Initiative. Qin Zhu and colleagues translated the scale into Chinese [23]; it comprises 23 items with a 3-point Likert scale (disagreement is 1 point, uncertainty is 2 points and agreement is 3 points) and is employed to evaluate the existing level of breastfeeding knowledge among nursing and other medical students in China, as well as to monitor and assess the outcomes of breastfeeding education. The CBKS includes 14, 17, and 22 items with reverse scoring. Scores on the scale range from 23 to 69, with higher scores indicating greater depth of breastfeeding knowledge. Validation of the tool was conducted among undergraduate nursing students in China, yielding Cronbach’s alpha and half reliability coefficients of 0.70 and 0.73, respectively. In this study, knowledge levels were categorized on the basis of whether participants scored below or equal to or above the average or median knowledge score [24], which was considered the threshold for determining insufficient or sufficient knowledge, respectively.

#### 2.3.3. Iowa Infant Feeding Attitude Scale (IIFAS)

The study assesses students’ attitudes toward breastfeeding via the Chinese version of the IIFAS. The IIFAS is a widely used tool internationally to measure attitudes toward breastfeeding. The Chinese version utilized in this research was translated by Yanru He [25] from Taiwan Province with proper authorization. It consists of 17 items, with 8 items focusing on breastfeeding receiving positive scores and 9 items on formula milk feeding receiving reverse scores. Responses are rated on a 5-point Likert scale: strongly disagree (1 point), disagree (2 points), no opinion (3 points), agree (4 points), and strongly agree (5 points). The total scores range from 17 (indicating positive attitudes toward formula milk feeding) to 85 (indicating positive attitudes toward breastfeeding). On the basis of the total score, attitudes are categorized as follows: positive breastfeeding attitudes (>70–85 points), neutral attitudes (49–69 points), and positive attitudes toward formula milk feeding (17–48 points) [26]. The Chinese version of the IIFAS has been validated among breastfeeding women in Taiwan Province and has good internal consistency, with a Cronbach’s alpha coefficient of 0.74.

### 2.4. Data Collection

The questionnaires were distributed via Questionnaire Star (https://www.wjx.cn/, accessed on 1 October 2024–10 January 2025), an online platform for questionnaire research in China. Data confidentiality was maintained via multiple mechanisms, including technical safeguards and strict management protocols [27]. We obtained written informed consent from the students. For those who were not present on site, we confirmed their consent via telephone and obtained verbal informed consent. Students meeting the inclusion criteria voluntarily accessed the questionnaire by scanning its QR code on Questionnaire Star after class. All the questions were mandatory, and incomplete responses prevented submission. To mitigate potential social desirability bias in the questionnaire survey, participants were informed that the survey was anonymous and that their responses would be kept strictly confidential. After collection, two investigators verified the questionnaire to rule out submitting the same options because this could be regarded as a failure to complete the questionnaire seriously and carefully. Ultimately, 874 questionnaires were obtained, with 121 deemed invalid and excluded, resulting in 753 valid questionnaires and an 86.2% return rate. The participant selection process is outlined in Figure 1.

### 2.5. Data Analysis

The study utilized nonparametric statistics because of the lack of a normal distribution of the dependent variables. Categorical variables are presented as frequencies and percentages (*n*, %), whereas continuous variables are presented as medians and percentile ranges (25th and 75th percentiles). Spearman correlation analysis was used to assess the correlation between knowledge and attitude scores. Univariate analysis, including the Mann–Whitney U test for two-group comparisons and the Kruskal–Wallis H test for multiple-group comparisons, was used to examine differences in IIFAS scores across various subgroups. Multiple linear regression (hierarchical and stepwise) was employed to identify key factors influencing breastfeeding attitudes among nursing undergraduates, with the assumption of noncollinearity met (variance inflation factor range: 1 to 1.668). Data analysis was conducted via the Statistical Package for the Social Sciences (SPSS Inc., Chicago, IL, USA), version 23.0, with statistical significance set at *p* < 0.05.

### 2.6. Ethical Considerations

The Anhui Medical University ethics committee approved the study protocol (permit number: 83230275). All students voluntarily completed the questionnaire. The personal information of all the students was kept confidential during the data collection and analysis.

## 3. Results

### 3.1. Correlations Between CBKS Scores and IIFAS Scores

Spearman correlation analysis revealed a significant positive correlation between knowledge scores and attitude scores (*r* = 0.462, *p* < 0.001). The participants’ CBKS scores were subsequently converted into binary variables on the basis of a median score of 54, with scores ranging from 27 to 54 classified as insufficient knowledge and scores ranging from 55 to 68 classified as sufficient knowledge. Detailed descriptive statistics of the sample are presented in Table 1.

### 3.2. Participants’ Demographic Characteristics and Comparison of IIFAS Scores

Participants aged 17 years to 23 years were included in the study. The Mann–Whitney U test and Kruskal–Wallis H test revealed no significant differences in IIFAS scores among the four factors: residential location type, being the only child, interest in major, and witnessing breastfeeding (*p* > 0.05). Conversely, significant differences were observed in the IIFAS scores for the remaining nine factors (*p* < 0.05). Further details are provided in Table 1.

### 3.3. The Current Status of Knowledge, Attitudes, and Classification of Infant Feeding Attitudes Are Shown in Table 2 and Table 3

The nursing undergraduates surveyed showed a range of 27–68 in breastfeeding knowledge scores and 29–84 in breastfeeding attitude scores. Only 21 students exhibited a positive attitude towards breastfeeding.

### 3.4. Stratified Analysis of Participants by IIFAS Classification

Stratified analysis based on IIFAS categories (Table 4) revealed that the proportion of students with a positive breastfeeding attitude was higher among the subgroups. Specifically, 12.9% (9/70) of midwifery students and 6.6% (17/257) of junior-year students held positive attitudes. This proportion was also higher among students with sufficient breastfeeding knowledge (5.2%, 17/330) and those who were very willing to breastfeed (10.7%, 6/56).

### 3.5. Hierarchical and Stepwise Multiple Regression Analysis

Multiple linear regression models (hierarchical regression and stepwise regression) were used to analyze the relationships between the IIFAS scores (dependent variables) and the 9 statistically significant factors from Table 1 (independent variables). We obtained a regression model with five statistically significant independent variables. The results revealed that CBKS scores (reference range 27–54, 55–68 (*β* = 2.975, *p* < 0.001)), grade (reference: freshman, junior (*β* = 2.887, *p* < 0.001)), major (reference: elderly service management, midwifery (*β* = 3.235, *p* < 0.001)), willingness to breastfeed in the future/support wife’s breastfeeding (reference: very reluctant, very willing (*β* = 8.089, *p* < 0.001)), willingness (*β* = 5.194, *p* < 0.001), uncertainty (*β* = 3.027, *p* < 0.001), and feeding types before six months (reference: other food feeding, formula milk feeding (*β* = −1.591, *p* = 0.020) were the primary influencing factors on IIFAS scores (*R*^2^ = 0.327, *F* = 51.666, *p* < 0.001). Further details can be found in Table 5.

## 4. Discussion

This study provides insight into the attitudes of nursing undergraduates in China toward breastfeeding and the factors influencing these attitudes. The results indicated that only a small number of students exhibited a positive attitude on breastfeeding, with the majority adopting a neutral stance on infant feeding. Notably, factors such as CBKS scores, grade, major, breastfeeding intention and feeding type before six months emerged as significant determinants of breastfeeding attitudes.

### 4.1. Infant Feeding Attitudes Among Nursing Undergraduates

The IIFAS total score in this study ranged from 29 to 84, with a median of 54 (51, 59), indicating a medium or lower-middle level, which was lower than the scores reported in other studies [26,28]. Furthermore, the well-documented insufficiency of breastfeeding knowledge among nursing students provides a fundamental explanation [29,30]. Without a solid evidence-based understanding, students lack the foundation to develop strong convictions, naturally leading to ambivalence. The neutral attitudes of nursing undergraduates toward breastfeeding may be influenced by a variety of realistic factors [31]. In China, despite the strong advocacy of breastfeeding in traditional culture and official campaigns, the competing demands in modern society, particularly the lack of workplace support for working mothers, generate conflicting social messages [32]. Students may perceive this gap between the ideal and reality, which in turn leads to a cautious “neutral” stance rather than firm endorsement.

Research has indicated that sociocultural background influences students’ breastfeeding attitudes [33]. For example, Syrian students favor breastfeeding, which may be attributed to cultural and religious factors. In contrast, Hungarian students tend to favor a more flexible approach, with some even showing a preference for formula feeding, which may be influenced by the Western practice of formula feeding [34]. Moreover, more positive attitudes toward breastfeeding have also been reported among Vietnamese medical students [35], which may be influenced by Southeast Asian cultural traditions. This difference may help explain the varying attitudes toward breastfeeding among Eastern and Western students.

Although the IIFAS was originally designed for breastfeeding women, its cross-population application may, to some extent, limit the scores of nursing undergraduates. However, some studies have also applied it to student populations and demonstrated good applicability [34,36]. The neutral scores reflect the genuine ambivalence of the nursing undergraduates, rather than a measurement artifact. This situation hinders nursing undergraduates from effectively supporting and promoting breastfeeding in China. Investigating the factors influencing students’ breastfeeding attitudes and implementing interventions to foster positive attitudes are crucial for enhancing breastfeeding support for pregnant women in the future.

### 4.2. Factors Associated with Nursing Undergraduates’ Attitudes Toward Breastfeeding

#### 4.2.1. Influence of CBKS Scores on Attitudes Toward Breastfeeding

Our study revealed a positive correlation between higher CBKS scores and improved IIFAS scores, suggesting a direct impact of knowledge levels on newborn feeding attitudes. This finding aligns with previous research among Portuguese medical students [36]. Future breastfeeding education programs should focus on designing effective teaching strategies and providing more breastfeeding information to improve the knowledge level of nursing undergraduates and promote the formation of positive attitudes [37]. Moreover, Agueda et al. showed that a teaching syllabus based on progressive transversal learning and participation in the real health environment of pregnant women could be an effective strategy to help students acquire more knowledge about breastfeeding [38]. We should create a real environment to cultivate positive attitudes among nursing undergraduates toward breastfeeding.

#### 4.2.2. Influence of Grade on Breastfeeding Attitudes

In this study, we found that senior students scored higher on the IIFAS than lower-grade students did. This contrasts with Vandewark et al.’s findings from a foreign study, where no significant differences in infant feeding attitudes were noted between lower-grade and graduating students [39]. Conversely, Jefferson et al. demonstrated significant variations in IIFAS scores among college students across different grades, with seniors exhibiting higher scores, which aligns with our results [40]. Juniors, having completed a substantial portion of their training program and acquired more knowledge, may exhibit more positive breastfeeding attitudes than freshmen and sophomores do. Furthermore, as students progress through their academic years, they may increasingly focus on career planning and practical applications, potentially influencing their breastfeeding attitudes and contributing to higher IIFAS scores. These findings underscore the importance of integrating breastfeeding-related coursework early in education and reinforcing and expanding this knowledge as students advance in their studies.

#### 4.2.3. Influence of Major on Breastfeeding Attitudes

Compared with the students majoring in elderly service management and nursing, the midwifery students reported higher scores in the IIFAS. Midwifery students receive more systematic and comprehensive breastfeeding education through dedicated courses each academic year and practical experience during obstetric or neonatology rotations. In contrast, nursing students primarily learn about breastfeeding in courses such as Obstetrics and Gynecology Nursing and Maternal and Child Nutrition (typically in the third year). Students majoring in elderly service management do not have any courses related to breastfeeding. The focused curriculum and hands-on clinical training in midwifery, which is directly linked to maternal and infant health, likely contribute to the active engagement and positive attitudes toward breastfeeding among midwifery students, reflected in their higher IIFAS scores. This outcome supports the effectiveness of our breastfeeding education intervention in shaping students’ professional attitudes, which is consistent with previous research findings [41,42].

In undergraduate nursing education in China, almost no school regards breastfeeding as a separate compulsory course. A majority of students are only exposed to knowledge of breastfeeding in several classes of obstetrics and gynecology nursing, and the course information obtained by students is seriously insufficient. Despite completing the specific mother–infant learning unit in the undergraduate curriculum, students often demonstrate inadequate breastfeeding knowledge, as revealed in Yang et al.’s study [29]. This underscores the necessity for enhanced breastfeeding education. Implementing innovative teaching approaches such as high-fidelity simulation and scenario-based teaching can facilitate students’ learning.

#### 4.2.4. Influence of Breastfeeding Intentions on Breastfeeding Attitudes

In this study, we also found that individuals with a strong willingness to breastfeed presented higher IIFAS scores than did those with significant reluctance. This finding highlights a positive correlation between nursing undergraduates’ breastfeeding attitudes and their intention to breastfeed. Previous research by Sara Moukarzel et al. revealed that students’ breastfeeding intentions are influenced by their level of knowledge [43]. Additionally, Celina Reyes et al. demonstrated that a single educational session may increase the intention of adolescent females to exclusively breastfeed in the future [28]. The enhancement of school-based breastfeeding education among nursing undergraduates is recommended to improve services for pregnant women. Furthermore, there is a need to focus on cultivating positive attitudes toward breastfeeding among nursing undergraduates within the academic curriculum. Therefore, establishing a supportive breastfeeding environment on campuses is crucial for fostering breastfeeding intentions among students during their studies [44].

#### 4.2.5. Influence of Feeding Type Before Six Months on Attitudes Toward Breastfeeding

Another relevant aspect to consider is personal experience, compared with other feeding types, formula milk consumption before six months was negatively correlated with the IIFAS score (*β* = −1.591, *p* = 0.020). This finding may be attributed to factors such as limited awareness of the benefits of breastfeeding [45], pervasive marketing of formula milk [46], and the influence of social and cultural background [47]. To address this issue, enhancing awareness of the benefits of breastfeeding through education and providing scientific information to correct misconceptions can help cultivate positive attitudes toward breastfeeding among students.

### 4.3. Implications for Nursing Education

A study conducted in the United States revealed that community-based breastfeeding peer support can increase local breastfeeding rates [48]. We can draw upon this model [49]. First, teachers provide professional knowledge to help students perceive the benefits of breastfeeding and correct any misconceptions they may hold. Second, as students learn about breastfeeding, they might face various pressures and challenges. In this context, support from the school and peers can offer emotional comfort [50], helping students maintain a positive mindset, thereby facilitating a shift from neutral to positive attitudes toward breastfeeding.

### 4.4. Limitations

Our study has several limitations. First, it was conducted solely at a medical college in Anhui Province, China, thus limiting the generalizability of the results to other regions within China and internationally. We observed disparities in sample sizes among students from the three majors, which is consistent with findings from previous studies [51]. The limited sample sizes may affect the reliability and generalizability of the results [52]. However, this discrepancy is attributable to practical constraints, particularly the limited enrollment quotas in midwifery and elderly service management majors, which made it challenging to obtain larger samples. Nevertheless, the issue of breastfeeding education in these two majors is worthy of attention. To some extent, small-sample majors can offer unique perspectives, facilitating the identification of specific educational needs and challenges, and thus hold significant value for teaching improvement [52,53]. Second, as a cross-sectional study, it focused solely on the current situation and factors influencing the attitudes of nursing undergraduates toward breastfeeding, without clarifying the pathways among these factors and their long-term impacts on attitudes. Third, this study did not explore other variables (such as potential self-reporting bias and the absence of socioeconomic or cultural variables) that could influence the attitudes of nursing undergraduates toward breastfeeding, thereby restricting a comprehensive understanding of the factors associated with their attitudes. Finally, although the CBKS adequately captured breastfeeding knowledge, its reliability could be improved.

### 4.5. Recommendations

To support breastfeeding success, we suggest prioritizing evidence-based nutritional interventions. This involves early incorporation of maternal nutrition and lactation modules into the curricula for non-midwifery majors with low IIFAS scores, emphasizing nutrients like omega-3s and lactation-specific micronutrients [54]. Implementing partner education programs with local hospitals can establish peer-led breastfeeding support initiatives for practical experience. Emphasizing the significance of teacher advocacy is important and offers a viable strategy for intervention by improving educators’ understanding of maternal nutrition during pregnancy and the postpartum period. Furthermore, promoting interdisciplinary collaborations between nutritionists and nurses to develop integrated training modules aligned with global Baby-Friendly Hospital Initiative standards is recommended.

Future research should also expand the sample size, geographical scope, consider factors influencing breastfeeding attitudes, conduct longitudinal studies with multi-center cohorts on the impact of nutrition education, and enhance the reliability of the IIFAS through psychometric analyses.

## 5. Conclusions

The predominantly neutral attitudes toward breastfeeding among Chinese nursing undergraduates highlight a critical gap in current education strategies. To effectively address this, curriculum reforms should target key modifiable factors—particularly knowledge building and personal intention—within nursing training. This evidence-based framework is adaptable across settings, yet realizing its full potential in different cultural contexts requires sustained institutional and policy commitment.

## Figures and Tables

**Figure 1 nutrients-17-03169-f001:**
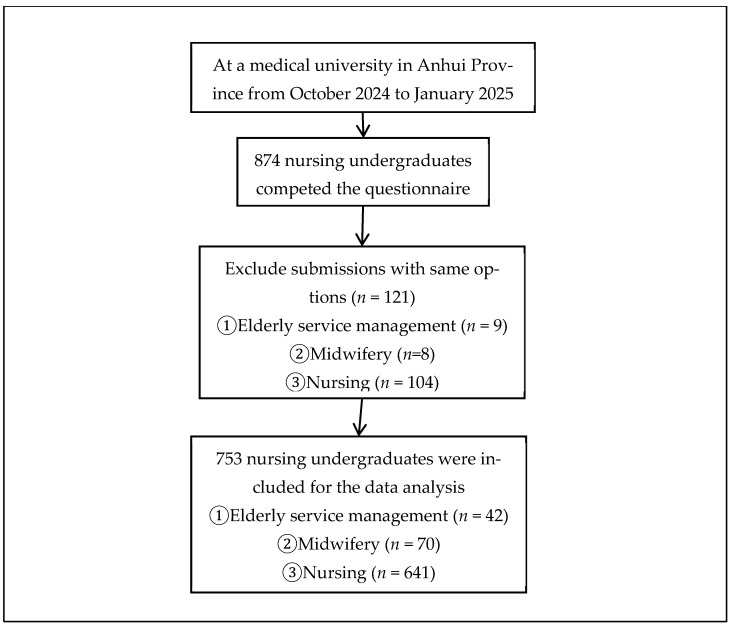
Flowchart of study participant enrollment.

**Table 1 nutrients-17-03169-t001:** Participants’ demographic characteristics and comparison of IIFAS scores (*n* = 753).

Variables	*n* (*n*%)	M (P25, P75)	Z/H Value	*p* Value
Gender			−2.641 ^a^	0.008
Male	166 (22.0%)	55 (52, 59)		
Female	587 (78.0%)	54 (50, 58)		
Residential location type			0.197 ^b^	0.978
Rural	299 (39.7%)	54 (51, 58)		
Township	92 (12.2%)	54 (52, 58)		
County town	181 (24.0%)	54 (50, 59)		
City	181 (24.0%)	54 (50.50, 58.50)		
The only child			−0.324 ^a^	0.746
Yes	128 (17.0%)	55 (51, 59)		
No	625 (83.0%)	54 (51, 58.50)		
Grade			46.553 ^b^	<0.001
Freshman	265 (35.2%)	54 (50, 57)		
Sophomore	231 (30.7%)	53 (49, 56)		
Junior	257 (34.1%)	56 (52, 61)		
Major			24.110 ^b^	<0.001
Elderly service management	42 (5.6%)	53.50 (51, 58.25)		
Midwifery	70 (9.3%)	59 (53, 64)		
Nursing	641 (85.1%)	54 (50, 58)		
Interest in major			5.426 ^b^	0.246
Very uninterested	8 (1.1%)	53 (48.25, 56.50)		
Uninterested	38 (5.0%)	53.50 (50, 57.25)		
Uncertain	346 (45.9%)	54 (50, 58)		
Be interested	333 (44.2%)	55 (51, 59.50)		
Extremely interested	28 (3.7%)	54 (50.50, 58.75)		
Mother-infant related courses			−7.354 ^a^	<0.001
Yes	271 (36.0%)	56 (52, 61)		
No	482 (64.0%)	53 (50, 57)		
Internship experience in obstetrics or neonatology			−4.696 ^a^	<0.001
Yes	167 (22.2%)	56 (52, 61)		
No	586 (77.8%)	54 (50, 58)		
Teachers advocate breastfeeding in class			−7.776 ^a^	<0.001
Yes	493 (65.5%)	56 (52, 60)		
No	260 (34.5%)	52 (49, 56)		
Breastfeeding intention			129.617 ^b^	<0.001
Very reluctant	72 (9.6%)	49.50 (44.25, 56)		
Unwilling	112 (14.9%)	51 (48, 54)		
Uncertain	257 (34.1%)	54 (51, 57)		
Be willing	256 (34.0%)	56 (53, 61)		
Very willing	56 (7.4%)	59.50 (54.25, 64)		
Witness the breastfeeding			−1.486 ^a^	0.137
Yes	603 (80.1%)	54 (51, 59)		
No	150 (19.9%)	54 (50, 58)		
Feeding types before six months			27.777 ^b^	<0.001
Exclusive breastfeeding	311 (41.3%)	55 (52, 59)		
Breast milk + formula milk feeding	345 (45.8%)	54 (50, 58)		
Formula milk feeding	74 (9.8%)	52 (48, 56)		
Other food feeding	23 (3.1%)	54 (51, 57)		
CBKS scores			−10.089 ^a^	<0.001
27–54	423 (56.2%)	53 (50, 56)		
55–68	330 (43.8%)	57 (53, 61)		

^a^ Mann-Whitney U test; ^b^ Kruskal–Wallis H-test; Other food feeding refers to rice flour, porridge and animal milk, etc.

**Table 2 nutrients-17-03169-t002:** Current situation of scoring on breastfeeding knowledge and attitude.

Items	Number of Entries	Score Range	Actual Total Score (P25, P75)	Minimum Score	Maximum Score
Knowledge	23	23–69	54 (49, 58)	27	68
Attitude	17	17–85	54 (51, 59)	29	84

**Table 3 nutrients-17-03169-t003:** Classification of infant feeding attitude.

Classification	IIFAS Scores	*n* (*n*%)
Positive breastfeeding attitude	70–85	21 (2.8%)
Neutral attitude	49–69	630 (83.7%)
Positive formula milk feeding attitude	17–48	102 (13.5%)

**Table 4 nutrients-17-03169-t004:** Stratified Analysis of Participants by IIFAS Classification.

Variables	Positive Breastfeeding Attitude (70–85)	Neutral Attitude (49–69)	Positive Formula Milk Feeding Attitude (17–48)	*p* Value
CBKS scores				<0.001
27–54 (423)	4 (19.0%)	339 (53.8%)	80 (78.4%)	
55–68 (330)	17 (81.0%)	291 (46.2%)	22 (21.6%)	
Grade				<0.001
Freshman (*n* = 265)	1 (4.8%)	233 (37.0%)	31 (30.4%)	
Sophomore (*n* = 231)	3 (14.3%)	175 (27.8%)	53 (52.0%)	
Junior (*n* = 257)	17 (81.0%)	222 (35.2%)	18 (17.6%)	
Major				<0.001
Elderly service management (*n* = 42)	0 (0.0%)	37 (5.9%)	5 (4.9%)	
Midwifery (*n* = 70)	9 (42.9%)	57 (9.0%)	4 (3.9%)	
Nursing (*n* = 641)	12 (57.1%)	536 (85.1%)	93 (91.2%)	
Breastfeeding intention				<0.001
Very reluctant (72)	0 (0.0%)	44 (7.0%)	28 (27.5%)	
Unwilling (112)	1 (4.8%)	78 (12.4%)	33 (32.4%)	
Uncertain (257)	2 (9.5%)	225 (35.7%)	30 (29.4%)	
Be willing (256)	12 (57.1%)	233 (37.0%)	11 (10.8%)	
Very willing (56)	6 (28.6%)	50 (7.9%)	0 (0.0%)	
Feeding types before six months				0.001
Exclusive breastfeeding (311)	8 (38.1%)	275 (43.7%)	28 (27.5%)	
Breast milk + formula milk feeding (345)	12 (57.1%)	282 (44.8%)	51 (50.0%)	
Formula milk feeding (74)	1 (4.8%)	52 (8.3%)	21 (20.6%)	
Other food feeding (23)	0 (0.0%)	21 (3.3%)	2 (2.0%)	

**Table 5 nutrients-17-03169-t005:** Hierarchical and stepwise multiple regression analysis of IIFAS scores.

Independent Variable	*β*	Standard Error	Beta	*t*	*p*	95%CI for *β*
Constants	48.899	0.457		107.075	0.000	48.002, 49.795
CBKS scores						
27–54	Reference					
55–68	2.975	0.439	0.220	6.781	<0.001	2.114, 3.837
Grade						
Freshman	Reference					
Junior	2.887	0.448	0.204	6.440	<0.001	2.007, 3.767
Major						
Elderly service management	Reference					
Midwifery	3.235	0.714	0.140	4.531	<0.001	1.833, 4.637
Breastfeeding intention						
Very reluctant	Reference					
Very willing	8.089	0.859	0.317	9.417	<0.001	6.403, 9.775
Be willing	5.194	0.549	0.367	9.457	<0.001	4.116, 6.273
Uncertain	3.027	0.536	0.214	5.644	<0.001	1.974, 4.080
Feeding types before six months						
Other food feeding	Reference					
Formula milk feeding	−1.591	0.684	−0.071	−2.325	0.020	−2.934, −0.247

Note: *R*^2^ = 0.327, F = 51.666, *p* < 0.001.

## Data Availability

The datasets used in the current study are available from the corresponding author upon reasonable request.

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
