# Peer review of "Breastfeeding Attitudes and Their Associated Factors Among Chinese Nursing Undergraduates: A Cross-Sectional Study"

_nutrients, 2025, doi:10.3390/nu17193169_

Round 1
Reviewer 1 Report
Comments and Suggestions for Authors
This manuscript by Liu et al aims to examine attitudes towards breastfeeding in undergraduate nursing students in China and determine possible factors that may show correlation with the determined attitudes. Students were evaluated using scales evaluating their breastfeeding knowledge and attitudes. The authors found that the majority of students demonstrated a neutral attitude towards breastfeeding and a minority showed a positive attitude, that their attitudes correlated with breastfeeding knowledge scores and that factors including year of training, major and willingness to breastfeed/support breastfeeding influenced attitudes towards breastfeeding. The authors concluded that these data would provide guidance towards designing interventions to cultivate breastfeeding advocacy.
The topic is interesting and important since education of healthcare providers regarding the positive effects of breastfeeding is certainly a priority. There are, however, several issues that need to be addressed in this manuscript, as described below.
Due to the presentation of the IIFAS scores as median with 25th and 75th percentiles, it is unclear which student characteristics trended towards high IIFAS scores of 70-85% or the higher end of the 49-69 score range. A discussion of these characteristics that result in these outliers should be added to the paper’s discussion/ results.
The authors should discuss the applicability of the IIFAS scale to healthcare providers as opposed to breastfeeding mothers and if this might be a cause of limitation.
Author Response
Reviewer 1:
- Due to the presentation of the IIFAS scores as median with 25th and 75th percentiles, it is unclear which student characteristics trended towards high IIFAS scores of 70-85% or the higher end of the 49-69 score range. A discussion of these characteristics that result in these outliers should be added to the paper’s discussion/ results.
Response: We express our gratitude to the editor for the time involved in reviewing the manuscript and the professional comment. Following your suggestion, we employed a stratified analysis to explore the characteristics of nursing Undergraduates within the two IIFAS score ranges: 49-69 and 70-85. And added table 4 and text for explanation. The specific contents are in lines 201 to 210, page number 8 to 9.
- The authors should discuss the applicability of the IIFAS scale to healthcare providers as opposed to breastfeeding mothers and if this might be a cause of limitation.
Response: We are grateful for your constructive feedback and agree with your viewpoint. To address this point, we have provided an explanation in the manuscript and supported it with relevant citations. Please find it on page 11, lines 254-258.
We are deeply grateful for your time and input, as this opportunity has allowed us to enhance our work. We sincerely hope that you will find the revised version of our manuscript satisfactory. We sincerely appreciate your valuable suggestion.
Sincerely
The authors
Reviewer 2 Report
Comments and Suggestions for Authors
I commend the authors on a capably conducted study of an important topic. I have a few recommendations for refinement.
- Regarding the significance of their study, the authors indicate the following: "The findings may offer important insights for developing breastfeeding education programs, shaping curricula, and implementing targeted interventions, thereby serving as a reference for enhancing the breastfeeding attitudes of nursing undergraduates. The methodology and identified factors also offer a model that can be replicated and applied to nursing education on a global scale." Perhaps add one more sentence about the value of discovering factors that facilitate and inhibit breastfeeding knowledge and attitudes, including sociodemographic variations that may be useful in the targeting of educational efforts among those within the population of study.
- Are there any theoretical insights or concepts that might help facilitate the interpretation of the results? This is not something I treat as an obligation of the authors, but it could be useful. By way of example, some have used a multi-theory model (MTM) of health behavior change to study breastfeeding: https://pmc.ncbi.nlm.nih.gov/articles/PMC11016141/. That's probably a bit more client-focused, but I'm curious if theory could lend a bit more clarity to a study that's already quite good. I find that theory, even if used sparingly, can sharpen the analysis.
- Do the small numbers for elderly service management and midwifery cause any concerns? Please say more about this. Those numerical differences vs. nursing students are quite pronounced. Still, I'm glad they're reported and included in the study. Small issue in Table 1: Elderly service management is capitalized, but other variables are not. This is not consistent with other tables.
- A bit more detail about why the results are so skewed toward neutral attitudes would be helpful. Is that a result of the economic and political context concerning women's labor force participation responsibilities despite government promotion? The limits of a purely or mostly cultural (messaging) approach to breastfeeding promotion could explain this. Structural factors such as workplace supports, or their absence, can be very influential. See these articles for related factors: https://www.mdpi.com/2227-9032/10/12/2444.
- There has been a good bit of research on breastfeeding peer support groups led by lactation specialists, such as Baby Cafes and similar options in the U.S. Recommending those might be in order. Community-based and peer support are often crucial: https://www.mdpi.com/2036-7503/16/4/91.
Well done with some room for refinement.
Author Response
Reviewer 2:
- Regarding the significance of their study, the authors indicate the following: "The findings may offer important insights for developing breastfeeding education programs, shaping curricula, and implementing targeted interventions, thereby serving as a reference for enhancing the breastfeeding attitudes of nursing undergraduates. The methodology and identified factors also offer a model that can be replicated and applied to nursing education on a global scale." Perhaps add one more sentence about the value of discovering factors that facilitate and inhibit breastfeeding knowledge and attitudes, including sociodemographic variations that may be useful in the targeting of educational efforts among those within the population of study.
Response: Thank you very much for such an important suggestion. And we agree with this comment. In accordance with your suggestion, we have added this text to explain the value of identifying factors that promote and inhibit knowledge and attitudes toward breastfeeding, including sociodemographic variables that could inform educational efforts for the study population. Please find it on page 2, lines 83-87.
- Are there any theoretical insights or concepts that might help facilitate the interpretation of the results? This is not something I treat as an obligation of the authors, but it could be useful. By way of example, some have used a multi-theory model (MTM) of health behavior change to study breastfeeding: https://pmc.ncbi.nlm.nih.gov/articles/PMC11016141/. That's probably a bit more client-focused, but I'm curious if theory could lend a bit more clarity to a study that's already quite good. I find that theory, even if used sparingly, can sharpen the analysis.
Response: Thank you for your helpful reminder. We agree with this point. As you pointed out, introducing the Multi-Theory Model (MTM) of health behavior change can make the content of this study clearer and sharpen the analysis. This section has been incorporated into the manuscript on pages 2, lines 71-76.
- Do the small numbers for elderly service management and midwifery cause any concerns? Please say more about this. Those numerical differences vs. nursing students are quite pronounced. Still, I'm glad they're reported and included in the study. Small issue in Table 1: Elderly service management is capitalized, but other variables are not. This is not consistent with other tables.
Response: Thank you for your professional advice. As you mentioned, there are disparities in sample sizes among the three majors. We have addressed and discussed this issue in the manuscript, specifically on page 13, lines 350–358.
Response: We are grateful for you identifying the error in Table 1, and we fully acknowledge that this was an oversight in our manuscript. Following your valuable feedback, we have amended the entry, changing “Elderly service management” to the correct format:“elderly service management”. Please find it in table 1.
- A bit more detail about why the results are so skewed toward neutral attitudes would be helpful. Is that a result of the economic and political context concerning women's labor force participation responsibilities despite government promotion? The limits of a purely or mostly cultural (messaging) approach to breastfeeding promotion could explain this. Structural factors such as workplace supports, or their absence, can be very influential. See these articles for related factors: https://www.mdpi.com/2227-9032/10/12/2444.
Response: Thank you for this crucial suggestion. You are correct. Our study did indeed lack a discussion on “why a tendency toward a neutral attitude can be beneficial.” At the same time, we recognize the validity of your point regarding the impact of economic, political, cultural, and structural determinants. In line with your recommendation, these two significant sections have now been incorporated into the manuscript, along with the corresponding citations. Please find it on page 11, lines 235 to 244.
- There has been a good bit of research on breastfeeding peer support groups led by lactation specialists, such as Baby Cafes and similar options in the U.S. Recommending those might be in order. Community-based and peer support are often crucial: https://www.mdpi.com/2036-7503/16/4/91.
Response: Thank you for your valuable advice. We agree that community- and peer-based support is beneficial for increasing breastfeeding rates. Furthermore, this model could be adapted for use in breastfeeding education for undergraduate nursing students. We have added this discussion to the manuscript on page 13, lines 337 to 345, hoping to provide a new intervention method for breastfeeding education.
We are deeply grateful for your time and input, as this opportunity has allowed us to enhance our work. We sincerely hope that you will find the revised version of our manuscript satisfactory. We sincerely appreciate your valuable suggestion.
Sincerely
The authors
Reviewer 3 Report
Comments and Suggestions for Authors
The manuscript addresses a highly relevant topic in public health, considering the impact of breastfeeding on maternal and child health indicators and the crucial role of future nursing professionals in its promotion. The study presents a methodological design that is appropriate for its objective and provides empirical evidence of significance for the Chinese context, with potential applicability at an international level.
TITLE
The title is clear, concise, and informative, accurately reflecting the study design and target population.
ABSTRACT
The abstract follows an appropriate structure and presents relevant statistical data that support the conclusions. However, it is overly dense in numerical results; therefore, it would be advisable to summarize the statistical findings and highlight the key results more prominently.
INTRODUCTION
The introduction appropriately situates the Chinese context by comparing national rates with global averages and emphasizes the importance of nursing students as future healthcare providers.
It provides a broad overview of the benefits of breastfeeding; however, the extensive enumeration of benefits (for mother, child, and family) is somewhat redundant and already well established in the literature.
The scientific gap (the lack of adequate breastfeeding education in undergraduate nursing curricula) only appears in the final lines. It is recommended that this gap be highlighted earlier in the introduction. Additionally, a linking statement between the “benefits of breastfeeding” and the “justification of the study” would strengthen the originality and relevance of the research.
METHODS
About this section, the following considerations are made:
- The cross-sectional design is appropriate for the exploratory objective.
- The sample size is robust, with the calculation well justified through multiple regression analysis.
- The characterization of the sample is limited to demographic and academic variables; socioeconomic or cultural factors, which may be relevant to attitudes toward breastfeeding, could have been included.
- The use of validated instruments strengthens the rigor of data collection.
- The use of the Chinese version of the CBKS is a positive aspect, as it is a previously translated and validated instrument. However, the reported Cronbach’s alpha (α = 0.70) indicates an acceptable, though not high, level of internal consistency. This value is sufficient for exploratory research, but it suggests the need for some caution when interpreting the results. It is recommended that the authors briefly discuss this limitation, emphasizing that while the CBKS adequately captures breastfeeding knowledge, future studies could benefit from additional psychometric analyses (e.g., item re-evaluation or application in different samples) in order to enhance the reliability of the scale.
- Ethical approval is appropriately mentioned.
- It would be valuable to address strategies used to control for potential social desirability bias, which is common in questionnaire-based studies.
- The authors should clarify the reference to the “three training areas (general nursing, maternal health nursing—midwifery—and elderly services management).” Are these different degree tracks/specializations within undergraduate nursing, or do they represent variations in clinical training experiences within the same undergraduate program? This point was not sufficiently clear.
RESULTS
The results are presented clearly, using appropriate statistical tests. The multivariate analysis successfully identified independent factors associated with attitudes. The high response rate strengthens the credibility of the study. However, confidence intervals were not reported in the regression analyses, which limits the interpretation of the precision of the coefficients.
DISCUSSION
The discussion presents several positive aspects, including the comparison of findings with the international literature, highlighting similarities and differences; the appropriate interpretation of the influence of factors such as year of study, training area, and intention to breastfeed; and the suggestion of pedagogical strategies such as simulation and scenario-based teaching.
Nevertheless, some areas for improvement are noted:
- There is redundancy between the results and the discussion, with repetition of certain numerical data.
- The comparative analysis with other cultural contexts is insufficient; for instance, it would be valuable to explore differences between students in Western and Asian countries.
- The interpretation of the “neutral attitude” category is rather superficial. It would be helpful to discuss whether this reflects a lack of knowledge, indifference, or genuine neutrality.
LIMITATIONS AND RECOMMENDATIONS
Regarding the limitations, the authors appropriately acknowledge that this was a cross-sectional and single-center study. However, the discussion could be expanded to include other limitations, such as potential self-report bias and the absence of socioeconomic or cultural variables.
CONCLUSIONS
The conclusion clearly synthesizes the main findings and appropriately establishes a direct link to educational and curricular practice. However, this section could be developed in a more critical and realistic manner by acknowledging that curricular change also depends on broader educational and institutional policies. It is recommended that the authors include a stronger final statement emphasizing the international relevance of the findings and their applicability across different cultural contexts.
FINAL SECTIONS
Informed consent was described as “oral.” It would be important to justify this choice, as written consent is generally preferred in some international academic contexts. It may also be useful to add a note on how data confidentiality was ensured when using the online platform (Questionnaire Star).
The references should be more up to date (within the last five years).
OVERALL ASSESSMENT
Major Revisions
The manuscript is solid and of good quality, but it requires important revisions and some methodological clarifications to reach the standard expected for international publication.
Congratulations on this valuable study!
Author Response
Reviewer 3:
- ABSTRACT: The abstract follows an appropriate structure and presents relevant statistical data that support the conclusions. However, it is overly dense in numerical results; therefore, it would be advisable to summarize the statistical findings and highlight the key results more prominently.
Response: Thank you for your suggestions. We agree with your viewpoint and have revised the numerical part of the abstract. Please find it on page 1, lines 21 to 27.
- INTRODUCTION: The introduction appropriately situates the Chinese context by comparing national rates with global averages and emphasizes the importance of nursing students as future healthcare providers.
â‘ It provides a broad overview of the benefits of breastfeeding; however, the extensive enumeration of benefits (for mother, child, and family) is somewhat redundant and already well established in the literature.
Response: Thank you for your suggestions. As you mentioned, the benefits of breastfeeding for children, mothers, and families are already well-established. Therefore, we have removed this section and summarized it in one sentence.Please find it on page 1, lines 37 to 38.
â‘¡ The scientific gap (the lack of adequate breastfeeding education in undergraduate nursing curricula) only appears in the final lines. It is recommended that this gap be highlighted earlier in the introduction.
Response: Thank you for your suggestions. We agree with your point that the scientific gap-namely, the lack of sufficient breastfeeding education in undergraduate nursing programs-was only mentioned in the last few lines. To address the problem, we have now moved this content forward. Please find it on page 2, lines 57 to 61.
â‘¢ Additionally, a linking statement between the “benefits of breastfeeding” and the “justification of the study” would strengthen the originality and relevance of the research.
Response: Thank you for your suggestions. We agree with your point that the statement linking‘the benefits of breastfeeding’and‘the rationale for the study’ does indeed enhance the originality and relevance of the research. We have updated this section accordingly. Please find it on page 2, lines 63 to 65.
- METHODS
â‘ The characterization of the sample is limited to demographic and academic variables; socioeconomic or cultural factors, which may be relevant to attitudes toward breastfeeding, could have been included.
Response: Thank you for your suggestions. Our sample characteristics indeed lack socioeconomic and cultural factors related to breastfeeding attitudes. This is a limitation of our study, and it has been added to the limitations section. Please find it on page 13, lines 362 to 363.
â‘¡ The use of the Chinese version of the CBKS is a positive aspect, as it is a previously translated and validated instrument. However, the reported Cronbach’s alpha (α= 0.70) indicates an acceptable, though not high, level of internal consistency. This value is sufficient for exploratory research, but it suggests the need for some caution when interpreting the results. It is recommended that the authors briefly discuss this limitation, emphasizing that while the CBKS adequately captures breastfeeding knowledge, future studies could benefit from additional psychometric analyses (e.g., item re-evaluation or application in different samples) in order to enhance the reliability of the scale.
Response: Thank you for your suggestions. The Cronbach’s alpha for the CBKS was 0.70, which is not sufficiently high. As per your suggestion, we have now included this in the limitations section. Please find it on page 13, lines 365 to 368.
â‘¢ Ethical approval is appropriately mentioned.
Response: Thank you for your kind reminder. We have updated the section regarding ethical approval. Please find it on page 6, lines 176 to 180.
â‘£ It would be valuable to address strategies used to control for potential social desirability bias, which is common in questionnaire-based studies.
Response: Thank you for your kind reminder. We agree with your viewpoint that questionnaire surveys are indeed prone to potential social desirability bias. During the actual survey process, we informed participants that the questionnaire would be completed anonymously and that their responses would be kept strictly confidential to minimize this bias.We have updated the content in the data collection section. Please find it on page 5, lines 154 to 156.
⑤ The authors should clarify the reference to the “three training areas (general nursing, maternal health nursing—midwifery—and elderly services management).” Are these different degree tracks/specializations within undergraduate nursing, or do they represent variations in clinical training experiences within the same undergraduate program? This point was not sufficiently clear.
Response: Thank you for pointing this out. Our description of the three majors was not sufficiently clear. These three majors all belong to the College of Nursing, but they are different majors. Please find it on page 4, lines 96.
- RESULTS:The results are presented clearly, using appropriate statistical tests. The multivariate analysis successfully identified independent factors associated with attitudes. The high response rate strengthens the credibility of the study. However, confidence intervals were not reported in the regression analyses, which limits the interpretation of the precision of the coefficients.
Response: Thank you for your suggestions. We have updated Table 5 by adding the βvalues and 95% confidence intervals. Please find it on page 10, lines 224 to 225.
- DISCUSSION
â‘ There is redundancy between the results and the discussion, with repetition of certain numerical data.
Response: Thank you for your suggestions. We have removed the redundant content between the“3.2. Participants’ Demographic Characteristics and Comparison of IIFAS Scores”section and the“Discussion“. Please find it on page 6, lines 190 to 194.
â‘¡ The comparative analysis with other cultural contexts is insufficient; for instance, it would be valuable to explore differences between students in Western and Asian countries.
Response: Thank you for your suggestions. We agree that exploring the differences between students from Western and Asian countries would be valuable. This content has now been added to Section 4.1, the Discussion section, with corresponding references cited for support. Please find it on page 11, lines 245 to 253.
â‘¢ The interpretation of the “neutral attitude” category is rather superficial. It would be helpful to discuss whether this reflects a lack of knowledge, indifference, or genuine neutrality.
Response: Thank you for your suggestions. We agree with your viewpoint. In the section “4.1. Infant Feeding Attitudes Among Nursing Undergraduates,” we have provided a more in-depth explanation of “neutral attitude.”With corresponding references cited for support. Please find it on page 11, lines 235 to 244.
- LIMITATIONS AND RECOMMENDATIONS: Regarding the limitations, the authors appropriately acknowledge that this was a cross-sectional and single-center study. However, the discussion could be expanded to include other limitations, such as potential self-report bias and the absence of socioeconomic or cultural variables.
Response: Thank you for your constructive suggestions. We have added content to the discussion section, such as potential self-report bias and the absence of socioeconomic or cultural variables. Please find it on page 13, lines 362 to 363.
- CONCLUSIONS: The conclusion clearly synthesizes the main findings and appropriately establishes a direct link to educational and curricular practice. However, this section could be developed in a more critical and realistic manner by acknowledging that curricular change also depends on broader educational and institutional policies. It is recommended that the authors include a stronger final statement emphasizing the international relevance of the findings and their applicability across different cultural contexts.
Response: Thank you for your constructive suggestions. We agree with your viewpoint. In the conclusion section, we have added that curriculum changes also depend on broader educational and institutional policies, and have emphasized the international relevance of the findings and their applicability across different cultural contexts. Please find it on page 14, lines 376 to 378.
- FINAL SECTIONS
â‘ Informed consent was described as “oral.” It would be important to justify this choice, as written consent is generally preferred in some international academic contexts.
Response: Thank you for pointing this out. We obtained written informed consent from the students. However, for those who were not present, we obtained their verbal informed consent via telephone. This content has been updated in the data collection section. Please find it on page 5, lines 149 to 151.
â‘¡ It may also be useful to add a note on how data confidentiality was ensured when using the online platform (Questionnaire Star).
Response: Thank you for pointing this out. We agree with your viewpoint. Content regarding how Questionnaire Star ensures data confidentiality has been added. This content has been updated in the data collection section. Please find it on page 5, lines 148 to 149.
â‘¢ The references should be more up to date (within the last five years).
Response: Thank you for pointing this out. Based on your feedback, we have updated some of the references. Please find it on page 14 to 17, lines 403 to 544.
We are deeply grateful for your time and input, as this opportunity has allowed us to enhance our work. We sincerely hope that you will find the revised version of our manuscript satisfactory. We sincerely appreciate your valuable suggestion.
Sincerely
The authors
Round 2
Reviewer 2 Report
Comments and Suggestions for Authors
I commend the authors on a sound and compelling revision.
The word "Specially" could be replaced with "Specifically" in line 74.
Also, I may have been unclear about table category capitalization. I think the journal typically wants the first letter of a variable name to be capitalized, so "Elderly service provision" was correct. I saw capitalization inconsistencies and didn't mean to suggest that all variables should be completely lower case in tables.
Author Response
Reviewer 2:
- The word "Specially" could be replaced with "Specifically" in line 74.
Response: Thank you for pointing this out. We agree with you and have changed “Specially” to “Specifically”. Please find it on page 2, lines 73.
- Also, I may have been unclear about table category capitalization. I think the journal typically wants the first letter of a variable name to be capitalized, so "Elderly service provision" was correct. I saw capitalization inconsistencies and didn't mean to suggest that all variables should be completely lower case in tables.
Response: Thank you very much for pointing this out. We agree with you that we have capitalized the first letter of the variable name. Please find it in Table 1 (lines 195-196), Table 4 (lines 208-209), and Table 5 (lines 223-224).
We are deeply grateful for your time and input, as this opportunity has allowed us to enhance our work. We sincerely hope that you will find the revised version of our manuscript satisfactory. We sincerely appreciate your valuable suggestion.
Sincerely
The authors